# Effect of a New Substance with Pyridoindole Structure on Adult Neurogenesis, Shape of Neurons, and Behavioral Outcomes in a Chronic Mild Stress Model in Rats

**DOI:** 10.3390/ijms25020845

**Published:** 2024-01-10

**Authors:** Alexandra Zvozilova, Alexandra Reichova, Mojmir Mach, Jan Bakos, Romana Koprdova

**Affiliations:** 1Institute of Experimental Pharmacology and Toxicology, Centre of Experimental Medicine, Slovak Academy of Sciences, 841 04 Bratislava, Slovakia; exfaalba@savba.sk (A.Z.); romana.koprdova@savba.sk (R.K.); 2Institute of Experimental Endocrinology, Biomedical Research Center, Slovak Academy of Sciences, 845 05 Bratislava, Slovakia; alexandra.reichova@savba.sk; 3Institute of Physiology, Faculty of Medicine, Comenius University, 811 08 Bratislava, Slovakia

**Keywords:** major depressive disorder, antidepressants, animal model, behavior, pyridoindoles, triple reuptake inhibitors, immunohistochemistry

## Abstract

Despite an accumulating number of studies, treatments for depression are currently insufficient. Therefore, the search for new substances with antidepressant potential is very important. In this study, we hypothesized that treatment with a newly synthesized pyridoindole derivative compound SMe1EC2M3 would result in protective and antidepressant-like effects on behavioral outcomes and reverse the impaired adult hippocampal neurogenesis caused by chronic mild stress (CMS). We found that chronic administration of 5 mg/kg and 25 mg/kg SMe1EC2M3 to adult Sprague Dawley rats ameliorated the consequences of CMS on immobility and swimming time in a forced swim test. A slight sedative effect of the highest dose of SMe1EC2M3 in the nonstress group was observed in the open field. SMe1EC2M3 in the highest dose ameliorated CMS-induced decreases in the sucrose preference test. Administration of SMe1EC2M3 significantly increased SOX2-positive cells in the hippocampal dentate gyrus (DG) in CMS compared to control animals. A significant reduction in glial fibrillary acid protein (GFAP)-positive cells in the DG of CMS compared to control animals was observed. Administration of both 5 and 25 mg/kg SMe1EC2M3 significantly increased signal of GFAP-positive cells in the DG of CMS animals. No such effects of SMe1EC2M3 were observed in the cornu ammonis hippocampal area. Additionally, we found that incubation of primary hippocampal neurons in the presence of 1.50 µM SMe1EC2M3 significantly stimulated the length of neurites. Overall, we found that the negative effects of CMS on depression-like behavior are partially reduced by the administration of SMe1EC2M3 and are associated with changes in hippocampal neurogenesis and neuronal differentiation. SMe1EC2M3 represents a potential drug candidate with positive neuroplastic effects and neurogenesis-associated effects in therapeutic approaches to depression.

## 1. Introduction

Major depressive disorder (MDD) is one of the most common psychiatric disorders and the leading cause of disability worldwide [1]. The etiology of MDD is not yet completely understood, but the monoamine theory of depression is highly supported by the antidepressant mechanism of action that modulates monoaminergic systems [2].

The most prescribed first-line antidepressants that increase the availability of monoamines in the synaptic cleft are selective serotonin (5-HT) reuptake inhibitors (SSRIs) [3], followed by serotonin and noradrenaline (NE) reuptake inhibitors (SNRIs). It has also been suggested that simultaneous action on all three monoamine systems shows higher clinical efficiency [4,5]. Antidepressants from the group of triple reuptake inhibitors (TRIs) are able to increase the availability of 5-HT, NE, and dopamine (DA) at the same time [6], which makes them promising candidates for more potent drugs in the treatment of depression. Nevertheless, despite increasing studies, the treatment of depression is currently insufficient. Moreover, preclinical and clinical studies point to a wide range of numerous adverse effects, withdrawal syndrome, or long onset of action [7,8]. The clinical efficacy of available drugs is limited, and the evidence shows that only 20–40% of patients show full remission and almost 15% of the disabled do not respond to any of the commercially available drugs [4,9].

Therefore, there is a constant need to look for new potential antidepressants with higher efficacy, earlier onset, and minimum harmful side effects. Following our previous studies and in an effort to understand how antidepressant effects are mediated in the brain, we have focused our attention in this work on the pyridoindole derivative compound (±)-cis ethyl 8-methoxy-6-methyl-3,4,4a,5,9bH-hexahydro-1H-pyrido[4,3-b]indole-2-carboxylate (SMe1EC2M3, Figure 1). SMe1EC2M3 was synthesized and originally developed at the Institute of Experimental Pharmacology and Toxicology, Slovak Academy of Sciences, as one of 82 pyridoindole derivatives.

More evidence has been pointing toward pyridoindole derivatives as a possible replacement of benzodiazepines in the treatment of anxiety disorders. Behavioral studies in rodents showed that pyridoindole analog SL651498 ([6-Fluoro-9-metyl-2-fenyl-4-(pyrolidin-1-yl-karbonyl)-2,9-dihydro-1H-pyrido[3,4-b]indol-1-one]) has anxiolytic-like activity similar to diazepam [10]. Similar findings of anxiolytic properties were observed in pyridoindole derivatives SMe1EC2 and SMe1M2, derivatives of the compound stobadine (STO). This molecule is an optic stereoisomer of the neuroleptic drug γ-carboline Carbidine^®^ and was studied in detail with no adverse effects [11]. The neuroprotective impact of SMe1EC2 was previously observed in the form of reduced hippocampal swelling during ischemic conditions and on hippocampal slices exposed to hypoxia/reduced glucose concentration by analysis of synaptic transmission in the CA1 region [12]. The neuroprotective effects of SMe1EC2 were demonstrated in a mouse model of acute head trauma, alongside its proven strong antiradical effect [13,14,15]. Among the derivatives mentioned above, SMe1EC2M3 was selected due to its superior properties based on in silico studies.

In our previous immunohistochemical study, after an acute administration of SMe1EC2M3, we observed an increase in c-Fos-positive cells as a marker for neuronal activity in the brain linked to the pathogenesis of depression, such as the central amygdaloid nucleus and the nucleus paraventricularis of the hypothalamus [16]. Moreover, our in vivo electrophysiology assessment indicated that SMe1EC2M3 might be a potent 5-HT, NE, and DA reuptake inhibitor, which is supported by significant decreased immobility and increased active swimming in forced swim tests [16,17].

In this study, we decided to expand our knowledge about the potential protective effects of SMe1EC2M3 on behavior, adult hippocampal neurogenesis, and the fate of neuronal progenitors in conditions of chronic unpredictable stress. A chronic mild stress (CMS) animal model of depression was selected as a widely accepted and reliable model with the presence of depression-like behavior. Several weeks of exposure to various types of stressors with a different intensity which cannot be predicted by the animals result in behavioral, neurochemical, and neuroanatomical changes parallel to the symptoms of MDD [18,19]. Those symptoms may then be reversed by chronic treatment with different groups of antidepressants, which seems to have a positive effect on hippocampal neurogenesis [20]. We have chosen to explore three distinct subregions within the hippocampus, namely the dentate gyrus, CA1, and CA3, as they exhibit a high capacity for neurogenesis, undergo remodeling during depression, and play a central role in the investigation and understanding of depression’s pathophysiology [21,22,23]. We also assessed depression-like behavior using the sucrose preference test to evaluate one of the main symptoms of depression, anhedonia; then, we used the open field test to investigate motor activity and exploration, and also the forced swim test to reveal motivation to survive in animals. Based on our previous studies, we hypothesized that treatment with SMe1EC2M3 in selected doses would result in a protective and antidepressant-like effect on behavioral outcomes and reverse the impaired adult hippocampal neurogenesis caused by CMS. To obtain a more comprehensive picture of SMe1EC2M3 exposure, we used primary hippocampal cell cultures to evaluate cell shape, neurite outgrowth, and dendritic arborization.

## 2. Results

We performed a battery of behavioral tests that covered a comprehensive analysis of anhedonia and depression-like behavior to test the antidepressant efficacy of the SMe1EC2M3 compound. A two-way ANOVA revealed a significant main effect of our conditions (stress vs. nonstress) on the sucrose preference test (*p* ≤ 0.05, F_(1,62)_ = 6.820), as well as a main effect of treatment (*p* ≤ 0.01, F_(2,62)_ = 5.158, Figure 2). Fisher’s LSD post hoc analysis revealed that vehicle-treated animals in stress conditions had a significantly lower sucrose intake compared to vehicle nonstress animals (*p* ≤ 0.05); therefore, the efficacy of the chronic mild stress model was proven. Fisher’s LSD post hoc analysis also showed that administration of 25 mg/kg SMe1EC2M3 in the stress group significantly changed sucrose preference (*p* ≤ 0.05) and thus normalized the effect of stress conditions.

In the open field test, a two-way ANOVA revealed a significant main effect of the conditions (stress vs. nonstress) on the distance traveled (*p* ≤ 0.01, F_(1,65)_ = 9.691, Figure 3), as well as the interaction of treatment and conditions (*p* ≤ 0.05, F_(2,65)_ = 3.185, Figure 3). Fisher’s LSD post hoc analysis revealed that animals exposed to a higher dose of SMe1EC2M3 traveled a shorter distance than vehicle-treated animals in nonstress conditions. A significantly decreased locomotor activity of stress animals regardless of treatment compared to animals treated with vehicle in nonstress conditions was observed.

In the forced swim test (FST), we found significant differences in the main effect of treatment (*p* ≤ 0.001, F_(2,64)_ = 13.14), conditions (stress vs. nonstress) (*p* ≤ 0.01, F_(1,64)_ = 10.58), and their interaction (*p* ≤ 0.001, F_(2,64)_ = 14.38) on immobility time. Fisher’s LSD post hoc analysis revealed that vehicle-treated animals in stress conditions spent more time in immobility compared to vehicle nonstress animals (*p* ≤ 0.001, Figure 4a). Administration of SMe1EC2M3 in both doses significantly reduced immobility time in stress animals (each dose *p* ≤ 0.001, Figure 4a). A two-way ANOVA also revealed a significant main effect of the treatment on the measured time spent swimming in the FST (*p* ≤ 0.05, F_(2,64)_ = 4.349, Figure 4b). Fisher’s LSD post hoc analysis revealed that administration of SMe1EC2M3 in both doses increased the swimming time (*p* ≤ 0.01 for dose 5 mg/kg SMe1EC2M3; *p* ≤ 0.05 for dose 25 mg/kg SMe1EC2M3, Figure 4b) in stress conditions.

New neurons are continuously generated in the adult brain and this generation occurs in the hippocampus, particularly in the subgranular zone (SGZ) of the hippocampal dentate gyrus. To identify impairments in adult hippocampal neurogenesis caused by CMS, we performed an immunohistochemical analysis of progenitors and neuroblasts markers, such as Sox2 and GFAP, and the neuronal density marker NeuN in the DG, CA1, and CA3 hippocampal regions (bregma −3.30 mm to −4.52 mm).

A two-way ANOVA of the data obtained from the analysis of SOX2+-positive cells in the DG (Figure 5a) revealed a significant difference (*p* ≤ 0.001) for the factors interaction (F_(2,186)_ = 14.67) and treatment (*p* ≤ 0.001, F_(2,186)_ = 7.837). Tukey’s post hoc test showed a significant 15% reduction in SOX2+ cells in the vehicle stress group (*p* ≤ 0.001) compared to the vehicle nonstress group. Administration of 25 mg/kg SMe1EC2M3 significantly increased SOX2+ cells by 23% in the stress group (*p* ≤ 0.001) compared to the stress vehicle group and thus normalized the quantity of progenitor cells compared to the nonstress vehicle group. Subsequently, we focused on the CA1 and CA3 regions (Figure 5b and Figure 6c) and the two-way ANOVA analysis revealed a significant difference only for the treatment factor (CA1—*p* ≤ 0.01, F_(2,90)_ = 6.572; CA3—*p* ≤ 0.01, F_(2,90)_ = 5.104). A slight significant 11% decrease in SOX2 progenitor cells in the CA1 hippocampal area in the 5 mg/kg SMe1EC2M3-treated nonstress group (*p* ≤ 0.05) compared to the nonstress vehicle group was found (Figure 5c). Similar trends were found in the CA3 region (Figure 5b), but a significant decrease was observed in the stress group after 5 mg/kg SMe1EC2M3 administration (*p* ≤ 0.05).

Consistently with the findings on SOX2 progenitor cells, a two-way ANOVA of GFAP+-positive cells analyzed in the DG (Figure 6a) revealed a significant difference for the following factors: interaction (*p* ≤ 0.001, F_(2,186)_ = 8.553), stress vs. nonstress condition (*p* ≤ 0.05, F_(1,186)_ = 4.289), and treatment (*p* ≤ 0.0001, F_(2,186)_ = 11.75). Tukey’s post hoc test showed a significant 13% reduction in GFAP+ cells in the vehicle stress group (*p* ≤ 0.001) compared to the vehicle nonstress group. Administration of 5 mg/kg SMe1EC2M3 significantly decreased GFAP+ cells by 14% in the nonstress (*p* ≤ 0.001) and by 9% in the stress group (*p* ≤ 0.05) compared to the nonstress vehicle group. A significant increase (10%) in the GFAP+ cells in the DG hippocampal area was observed in the 25 mg/kg SMe1EC2M3 stress group compared to the stress vehicle group. Therefore, we confirmed that the decrease in progenitor markers (SOX2 and GFAP) in the DG hippocampal regions of adult Sprague Dawley rat brains after CMS was rescued after SMe1EC2M3 treatment. Statistical analysis revealed no change in GFAP+ cells in the CA1 and CA3 regions in stress conditions or after the treatment. No statistically significant changes in the number of NeuN+ positive cells (Figure 7) in stress conditions or after the treatment were observed in the hippocampal regions.

Primary neuronal cultures originating from various brain regions, including the hippocampus, are frequently used for studying the impact of diverse compounds on neurite outgrowth [24,25]. Therefore, the morphology of primary hippocampal neurons isolated from Wistar rat neonates (the day of birth, P0) was evaluated. After dissection, primary hippocampal cultures were cultivated for 7 days in a growing medium for the purpose of measurement of the longest neurites and neurite branching in immature neurons. At day in vitro (DIV) 7, cultures were treated either with or without SMe1EC2M3 (0.25; 0.50; 1.00; 1.50 µM) or ATRA (10 µM) for 48 h. The exposure of primary neurons to a positive control, ATRA, was used to examine neurite outgrowth. Statistical analysis of the average lengths of the longest neurites (Figure 8) by ANOVA revealed significant differences between the groups (*p* ≤ 0.0001, F_(5,742)_ = 5.809). Firstly, we compared neurite lengths between the control (CTRL) group and neurons after 48 h incubation with the positive control ATRA (Figure 8a) and observed a significant 20% increase (*p* ≤ 0.01) in the length of neurites. Afterward, we examined the effects of different concentrations of SMe1EC2M3 treatment. A Tukey–Kramer post hoc test revealed that only incubation in the presence of 1.50 µM SMe1EC2M3 significantly stimulated the length of neurites in neurons isolated from P0 Wistar rats by 13% (*p* ≤ 0.05; Figure 8a). We also analyzed the number of neurons with the longest neurite expressed as a percentage of the total measured neurons. Based on the length of the neurites, we divided the neurons into five categories for each experimental group (Figure 8b). We similarly found a higher percentage of neurons with the longest neurite over 200 µm in the neurons incubated with 1.50 µM SMe1EC2M3 (22.2%) and ATRA (31.9%) when compared to the CTRL untreated neurons (17.7%). However, no markable effects were observed after the application of other concentrations of the SMe1EC2M3 compound when compared to the CTRL group. By using Sholl analysis, we were able to obtain a detailed look at neurite length and the number of neurite branches in neurons in the visual field (Figure 9). We found a higher sum of intersections in the interval from 100 to 200 μm away from the nucleus in the ATRA positive control group (*p* ≤ 0.01) in comparison to the WT neurons. Overall, in these primary cell cultures, we observed an increase in neurite outgrowth after SMe1EC2M3 treatment; however, there were no statistically significant changes.

## 3. Discussion

We found that the negative effects of chronic mild stress on behavioral outcomes are partially reduced by the administration of SMe1EC2M3, a newly synthesized pyridoindole derivative compound. At the same time, immunohistochemical changes in markers of neurogenesis in the hippocampus in response to chronic mild stress were modulated by the administration of SMe1EC2M3, especially at a higher dose. At the level of primary hippocampal neuronal cultures incubated in the presence of SMe1EC2M3, we observed a prolongation of neurite outgrowth, which further complements the positive neuroplastic effects of this new substance.

A significant new finding of this study is that SMe1EC2M3 positively affects some elements of depression-like behavior in conditions of chronic mild stress. In particular, the significant effect of chronic SMe1EC2M3 administration on the reduction in immobility time in the forced swim test in stressed animals corresponds to the effects of other new and traditional substances suppressing the manifestations of depression [26,27]. Although some current studies question the validity of these behavioral tests, especially preclinical model of despair [28,29], our results were replicated in both control and stress-induced conditions; therefore, we consider them convincing. Moreover, we have already demonstrated the potential of acute SMe1EC2M3 administration to influence depression-like behavior in our previous study [16]; therefore, the present results reinforce the antidepressant efficacy of this novel compound in a stress model. However, the reduction in motor activity of control animals in the open field after the administration of higher doses of SMe1EC2M3 remains unexplained, suggesting a slight sedative effect of this substance in higher doses. The effectiveness of chronic antidepressant treatment in reversing or preventing CMS-induced decreases in intake or preference for sweet solutions has been described previously [30]. This is consistent with the present results insofar as the compound also slightly affected anhedonia in the sucrose preference test. It should be taken into consideration that such an effect always develops slowly, over a period of several weeks of treatment, and is always selective to CMS-treated animals, meaning that antidepressants do not increase the intake of or preference for sweet solutions in nonstress control groups [30]. We think that the short duration of exposure to SMe1EC2M3 might be the reason for only a moderate prevention of CMS-induced decrease in the intake of sweet solutions. Further and more comprehensive behavioral studies with longer exposure to obtain representative data are needed for the evaluation of SMe1EC2M3’s efficacy potential.

The consequences of chronic mild stress on the formation of hippocampal neuronal precursors and the differentiation of neuronal cells are known to some extent [31,32]. This study expands the known knowledge with new results of a decrease in the marker of Sox2 neural progenitor cells in the DG in response to chronic mild stress. Moreover, we confirmed a reduction in GFAP immunohistochemical signals in the DG after chronic mild stress. This is in line with previous results observing that chronic mild stress decreased the quantity of astrocytes [31]. It is somewhat surprising that we did not observe differences in NeuN signals, which are considered a marker of differentiating and terminally differentiated neurons, in any hippocampal area, either in response to the stress scenario or to SMe1EC2M3 treatment. Some previous studies observed a chronic mild stress-induced decrease in cytogenesis in the hippocampal DG and also the action of antidepressants [33,34]. It is also necessary to note a limitation of this study, which is that we did not investigate the proliferative effect of SMe1EC2M3 in the subventricular zone, as the main focus was on the hippocampus. However, it is clear that the issue of neuronal differentiation and survival and also the variability of postmitotic neurons in response to various types of stress and the effects of antidepressants [35] is broader and cannot be covered in this study.

The finding that a decrease in Sox2 and GFAP in the hippocampus is compensated for by the administration of SMe1EC2M3 is unique and needs further attention. In previous studies, it was shown that different subtypes of GFAP-expressing neural progenitors originate from the DG, which then migrate and form the granule cell layer in the developing hippocampus [36]. Changes in the interactions of glial neuronal cells with neurons in the adult brain, especially under repeated exposure to unpredictable chronic mild stress, may be related to the behavioral manifestations of depression [37].

The effect of SMe1EC2M3 on specific neurotransmitter systems—NE or 5-HT—remains unclear. Previous studies have confirmed that proliferating neural precursor cells and neurogenesis in the hippocampus are affected by stimulating NE neurotransmission [38], whereas stimulating 5-HT levels in the brain of adult experimental animals probably has the opposite effect [39]. The possible involvement of 5-HT signaling in brain development, neuronal differentiation, and neurite outgrowth has been more extensively investigated than its changes in the adult brain in response to the effects of antidepressants [40]. In this context, the finding that the incubation of hippocampal neuronal cells in the presence of SMe1EC2M3 led to neurite elongation is interesting, although this could be a short-term and possibly temporary effect. Various studies have demonstrated the stimulatory effect of antidepressants, or substances that have been shown to influence depression-like behavior, on neurite outgrowth in PC12 and SH-SY5Y neuronal cells, but also in primary hippocampal neurons [41,42,43]. In this context, it is crucial to further explore how SMe1EC2M3 affects the molecular processes of neuritogenesis. Given that SMe1EC2M3 is a prospective triple 5-HT, NE, and DA reuptake inhibitor with antidepressant-like properties [16], it is possible that its effects are manifested on the morphology and functions of neurons in various brain areas. It is important to mention that multitarget antidepressants that would affect multiple neurotransmitter systems such as NE, 5-HT, and DA release may potentially have an enhanced efficacy compared to monotherapy. This broader impact on neurotransmitter release allows for modulation across various neural pathways, potentially resulting in a more comprehensive approach to managing depression. The in vitro morphological changes in hippocampal neuronal cells observed in the presence of SMe1EC2M3 need to be verified in vivo in specific hippocampal areas or other relevant brain regions.

In conclusion, our results show that treatment with SMe1EC2M3 can be effective in ameliorating some depression-like symptoms observed in an animal model. Moreover, the present data suggest that SMe1EC2M3 is linked to changes in hippocampal neurogenesis and neuronal differentiation in stressful conditions. Overall, SMe1EC2M3 represents a potential drug candidate with neurogenesis-associated effects in therapeutic approaches to depression.

## 4. Materials and Methods

### 4.1. Animals

Male Sprague Dawley rats (initial weight 200–220 g, 3 months old) used in this study were acquired from Velaz (Prague, Czech Republic). All animals were housed (4 animals/cage; 38 × 59 × 25 cm large cage) under standard laboratory conditions (temperature: 22 ± 2 °C, humidity: 55 ± 10%) with a 12 h light/12 h dark cycle (lights on at 7 a.m.). Pelleted food and tap water were available ad libitum. The animals were handled and weighed regularly. All performed experiments were conducted in compliance with the Principles of Laboratory Animal Care issued by the Ethical Committee of IEPT, CEM SAS, and the experimental design was approved by the Animal Health and Animal Welfare Division of the State Veterinary and Food Administration of the Slovak Republic (Permit number Ro 1947-3/2020-220). For the behavioral study, 12 animals per group were used (n = 72) and subsequently, 6 animals/group were randomly chosen and transcardially perfused for the immunohistochemical analysis. The rats were anesthetized by a combined treatment of Zoletil (30 mg/kg, Virbac, Carros, France) and Xylariem (15 mg/kg, Riemser, Greifswald, Germany) in the volumes of 0.1 mL and 0.24 mL/300 g b.w., respectively. Then, they were perfused transcardially with 50 mL of cold isotonic saline containing 150 µL of heparin (5000 IU/mL, Zentiva, Bratislava, Slovakia) followed by 200 mL of fixative containing 4% paraformaldehyde (Sigma-Aldrich, Darmstadt, Germany) in 0.1 M phosphate buffer (pH 7.4).

### 4.2. Experimental Design

The rats were acclimatized in the animal housing facility for 1 week prior to the experimental procedures. The animals were divided into 6 experimental groups: vehicle nonstress, SMe1EC2M3 5 mg/kg nonstress, SMe1EC2M3 25 mg/kg nonstress; vehicle stress, SMe1EC2M3 5 mg/kg stress, SMe1EC2M3 25 mg/kg stress. The experimental animals in the stress groups were exposed to a particular stressor each day. The following types of stressors were used: overcrowding (9 instead of 4 rats were placed into one T4 cage (590 × 380 × 200 mm of size) for 12 h), air puff (air noise divided into 45 episodes, each of them lasting 1 min, randomized by a computer for 12 h), wet bedding (1000 mL of water poured into the cage for 12 h), predator stress (5 × 5 × 10 cm large perforated box filled with cat litter box content was put into the home cage for 12 h to provide a cat odor), food deprivation (food withdrawal for 12 h), and home cage tilted in 45 degree angle for 12 h (Table 1). The animals were exposed to the stressors for 3 consecutive weeks (Figure 10). From the 8th day of the CMS procedure, every day (12 a.m.) the animals were intraperitoneally (i.p.) injected with vehicle (sterile water for injection) or SMe1EC2M3 dissolved in vehicle (based on the previous study [16]) at the doses of 5 and 25 mg/kg for 2 weeks. The injection volume was 1 mL/kg body weight. The tested compound SMe1EC2M3 was synthesized in the Faculty of Natural Sciences, Comenius University in Bratislava, Slovakia. The purity of the compound was greater than 95% as determined by ^1^H-NMR analysis.

### 4.3. Behavioral Analysis

The behavioral tests were based on our previous studies [16]. In all tests, the movements of the rats were tracked with a digital camera and analyzed using the computer software ANYMAZE™ 6.1 (Stoelting Europe, Ireland). All experiments were conducted between 8 a.m. and 12 a.m. during the light phase; the intervals between the tests were 2 days. All parts of the apparatus were cleaned right after each animal with 60% ethanol to remove odor cues.

### 4.4. Sucrose Preference Test (SPT)

The sucrose preference test was used to detect anhedonia as the main symptom of MDD in rats. Firstly, all animals were habituated to the presence of 1% sucrose solution. We offered them two drinking bottles (one containing 1% sucrose and the other one tap water) for 24 h in their home cages. The position of the two bottles was switched every 2 h within 12 h to reduce any confounding produced by a side bias. On the 2nd day, we removed the sucrose bottle at 8 am and also food and water at 8 pm. Following this acclimation and food/water deprivation, on the 3rd (experimental) day, the individually housed animals had a free choice of either drinking 1% sucrose solution or regular water for 2 h (8 a.m.–10 a.m.). Sucrose preference was calculated as a percentage of the volume of sucrose intake over the total volume of water intake according to the formula: total volume of sucrose solution intake/total volume of all liquid intake (water + sucrose solution) × 100.

### 4.5. Forced Swim Test (FST)

For the forced swim test, the rats were placed in a glass cylinder (45 cm tall and 25 cm in diameter) filled with water (24 ± 1 °C) for a 15 min pretest period to induce depression-like behavior. The following experimental 5 min test started 24 h later. The depth of the water (30 cm) was sufficient to ensure that the animals could not touch the bottom of the container with their hind paws. All animals were returned to their home cage after resting under a heating lamp until dry. The time of immobility, climbing, and swimming of each animal was scored manually. Immobility (passive behavior) was defined as floating behavior without any movements other than those necessary to balance the body and keep the head above the water [44]. Escape-directed (active) behaviors were scored separately as vertical movement of the forepaws (climbing) or horizontal movement throughout the swim chamber (swimming).

### 4.6. Open Field (OF)

The open field test took place in a dark polyvinyl plastic arena illuminated with 100–200 lux, measuring 60 × 60 cm and surrounded by 25 cm high walls. Each session started by placing a rat in the center of the area and letting it freely explore the new environment for 5 min.

### 4.7. Immunohistochemistry

To analyze the phenotype of newborn cell populations and to determine adult neurogenesis in the hippocampal regions DG, CA1, and CA3 (bregma −3.30 mm to −4.52 mm) of adult Sprague Dawley rats, we investigated stage-specific molecular markers such as sex-determining region Y-box 2 (SOX2), neuronal nuclear antigen (NeuN), and glial fibrillary acid protein (GFAP) (n = 6 animals/group). Sex-determining region Y-box 2 (SOX2) has a key role in the regulation of neuronal progenitor cell proliferation in the central nervous system [45]. Neuronal nuclear antigen (NeuN) is a protein localized in the nuclei and perinuclear cytoplasm of neurons in the central nervous system. Also, it is a marker of neuronal maturation in the early human fetal nervous system, and it is widely used to assess the functional state of neurons [46]. Glial fibrillary acidic protein (GFAP) is used as a marker of mature astrocytes but is also expressed in radial glia-like cells in the adult forebrain [47].

After perfusion with 4% paraformaldehyde, brains were stored overnight in the same fixative and then sectioned on a 7000smz-2 Vibrotome (Campden Instruments Ltd., Leicestershire, UK) at a thickness of 50 µm. Sections were temporarily stored (4 °C) in 24-well plates in PBS + 0.01% sodium azide. For immunohistochemistry, randomly selected free-floating sections were blocked with 3% normal goat serum (NGS), 2% BSA, and 0.1% Triton X-100 in PBS. After 1 h (RT), sections were incubated with a combination of primary antibodies (Table 2) in 1% NGS, 1% BSA, and 0.4% Triton X-100 diluted in PBS overnight at 4 °C. After washing with cold PBS (5 min/3 times), the sections were incubated with corresponding secondary antibodies (Table 3) diluted in PBS for 1.5 h at room temperature. Subsequent sections were rinsed with PBS (5 min/3 times), and 300 nM DAPI was added to stain the nuclei. After final PBS washing, sections were mounted on glass slides with Fluoromount-G (Sigma-Aldrich, Darmstadt, Germany). Eight 50 µm thick sections of bilateral hippocampus/group were imaged using an Apotome2/Axio Imager.Z2 microscope with imagine device Axiocam 506 (Zeiss, Oberkochen, Germany) equipped with Ex./Em. filters (335–383/420–470; 455–495/505–555; 538–562/570–640) at 20× high magnification (EC Plan-Neofluar objective 20×, numerical aperture 0.5) with 3 µm Z-step intervals. The quantification of SOX2+ and GFAP+-positive cells was performed using Fiji/ImageJ software 2.15.0 and that of NEUN+-positive cells using QuPath0.3 software. In brief, positive cells were detected by QuPath0.3 software or manually counted by two different researchers from three different areas of interest in each part of the hippocampus (ROI [1 × 10^5^ µm^2^]/each DG, CA1, CA3 region of the bilateral hippocampus).

### 4.8. Preparation of Primary Hippocampal Neurons

Pregnant Wistar rats were obtained from Charles River Laboratories (Erkrath, Germany). The State Veterinary and Food Administration of the Slovak Republic approved all experimental procedures in accordance with the relevant legislation. Neonatal rats at postnatal day 0 (P0) were decapitated and the brains (n = 6 pups) were dissected in ice-cold Hank’s Balanced Salt Solution—HBSS (137 mM NaCl; 5.4 mM KCl; 0.5 mM MgCl_2_ × 6H_2_O; 0.4 mM MgCl_2_ × 6H_2_O; 0.44 mM KH_2_PO_4_; 0.34 mM Na_2_HPO_4_ × 7H_2_O; 1.25 mM CaCl_2_; 5.5 mM d-glucose) supplemented with 100 U/mL penicillin, 100 U/mL streptomycin, and 0.3 M Hepes (Sigma-Aldrich, Darmstadt, Germany) under a stereo microscope in order to collect the hippocampi. The isolation of primary hippocampal neurons was performed following a protocol from Reichova et al. [24,48]. Hippocampal tissues were dissociated for 20 min at 37 °C with an enzymatic solution (HBSS, 0.1% Trypsin, 0.1 mg/mL DNAse I). Cells were resuspended and plated on 24-well plates containing coverslips coated with 10 µg/mL poly-D-lysine (Sigma-Aldrich, Darmstadt, Germany) at a density of 0.8 × 10^5^/mL in RPMI medium (Sigma-Aldrich, Darmstadt, Germany) containing 10% fetal bovine serum. After 3 h of cell plating in a 37 °C and 5% CO_2_ incubator, the medium was replaced with a selective growing medium: Neurobasal A (Gibco, Billings, MT, USA), 100 U/mL penicillin and 100 U/mL streptomycin, 2 mM L-glutamine (Gibco, Billings, MT, USA) and 2% supplement B27 (Invitrogen, Waltham, MA, USA); DIV1 (day in vitro 1). After 5 days in vitro (DIV5), 50% of the growing media were exchanged. For the purpose of the evaluation of neurite outgrowth and dendritic arborization, different concentrations of SMe1EC2M3 (0.25; 0.50; 1.00; and 1.50 µM) or 10 µM all-trans retinoic acid (ATRA) (Sigma-Aldrich, Darmstadt, Germany) were added to the medium at DIV7 and DIV8 (48 h treatment).

### 4.9. Immunocytochemistry

After 48 h incubation (DIV9), the medium was removed and primary hippocampal cells were fixed with 4% paraformaldehyde, pH 7.4, for 20 min at room temperature (RT). The coverslips were washed 2 times with cold PBS and blocked in 3% (*v*/*v*) normal goat serum (NGS) and 0.1% Triton X-100 for 30 min at RT. Cells were stained with primary antibodies diluted in PBS with 3% serum, 0.1% Triton X-100 (Table 2), during 2 h at RT. Afterwards, the coverslips were rinsed 3 times with cold PBS and incubated with the corresponding fluorescent secondary antibodies diluted in PBS (Table 3) for 1 h at RT. The nuclei were stained with 300 nM DAPI (4′,6-diamidino-2-phenylindole; Thermo Fisher Scientific, Bratislava, Slovakia) for 1 min. The coverslips were mounted on glass slides with Fluoromount-G (Sigma-Aldrich, Darmstadt, Germany).

MAP2 (microtubule-associated protein 2)-positive cells were considered neurons. Image capturing was performed using an Apotome2 microscope (Zeiss, Oberkochen, Germany) at high magnification (20× objective, numerical aperture 0.5). Three coverslips per experimental group and at least 7 areas of interest per coverslip were evaluated using Fiji/ImageJ software 2.15.0. The arborization of dendritic trees was assessed using the Sholl analysis plugin [49]. The calculation of the number of intersecting dendrites with concentric circles in the interval from the soma up to 200 µm was performed. The length of the longest neurite was quantified from the edge of the nucleus to the apical end of the neurite by three independent members of the research team by manual tracing.

### 4.10. Statistical Analyses

Data analyses were performed using SigmaPlot 10 or GraphPad Prism 8.3 software. Outliers were identified and excluded using GraphPad PRISM’s ROUT method. Behavioral data are represented by mean ± SEM and were analyzed by two-way analyses of variance (ANOVA) with treatment and conditions (stress vs. nonstress) as main factors followed by Fisher’s LSD post hoc test, if applicable. Data analyses for immunocytochemical and immunohistochemical experiments were analyzed by one-way ANOVA for SMe1EC2M3 treatment (in vitro experiments) or two-way ANOVA with conditions (stress vs. nonstress) and treatment as main factors. As a post hoc, a Tukey–Kramer test for unequal sample sizes was used. Results are expressed as means ± SEM and a value of *p* ≤ 0.05 was considered statistically significant.

## Figures and Tables

**Figure 1 ijms-25-00845-f001:**
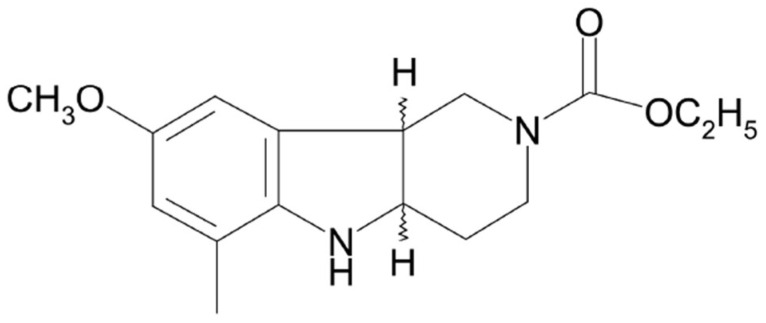
Structural formula of (±)-*cis* ethyl 8-methoxy-6-methyl-3,4,4a,5,9b*H*-hexahydro-1*H*pyrido[4,3-b]indole-2-carboxylate (SMe1EC2M3).

**Figure 2 ijms-25-00845-f002:**
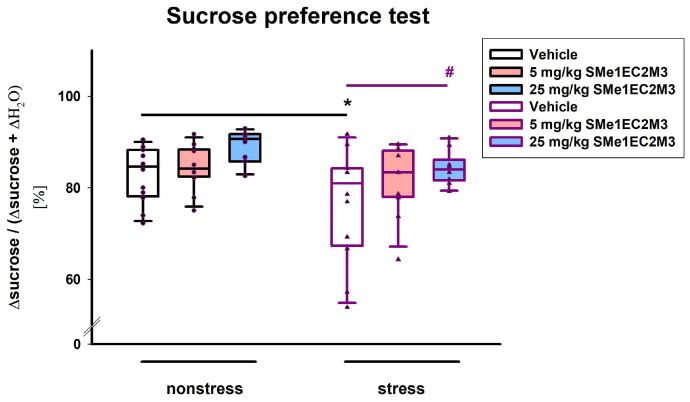
The effect of SMe1EC2M3 on sucrose preference. Means are represented in box plots ± SEM (n = 10–12 animals/group). Two-way ANOVA revealed main effect of stress vs. nonstress conditions (*p* ≤ 0.05) and main effect of treatment (*p* ≤ 0.01). Fisher’s LSD post hoc test revealed significantly different values, marked with * *p* ≤ 0.05, compared to nonstress vehicle group, and with ^#^ *p* ≤ 0.05 compared to stress vehicle group.

**Figure 3 ijms-25-00845-f003:**
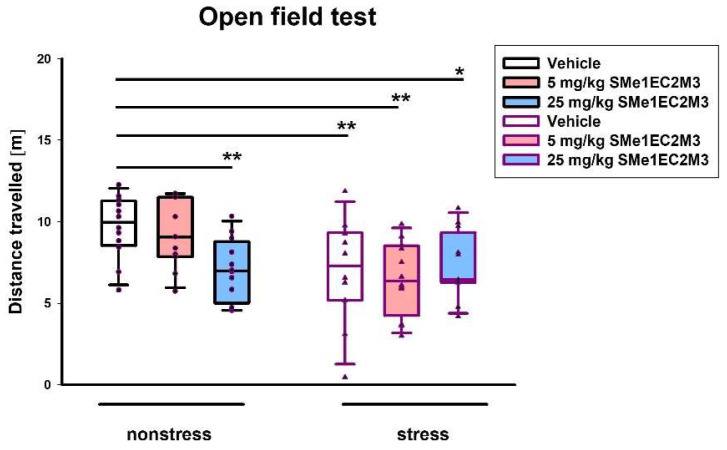
The effect of SMe1EC2M3 on distance traveled in open field test. Means are represented in box plots ± SEM (n = 11–12 animals/group). Two-way ANOVA revealed main effect of stress vs. nonstress conditions (*p* ≤ 0.01) and an interaction of treatment and conditions (*p* ≤ 0.05). Fisher’s LSD post hoc test revealed significantly different values, marked with * *p* ≤ 0.05; ** *p* ≤ 0.01 compared to nonstress vehicle group.

**Figure 4 ijms-25-00845-f004:**
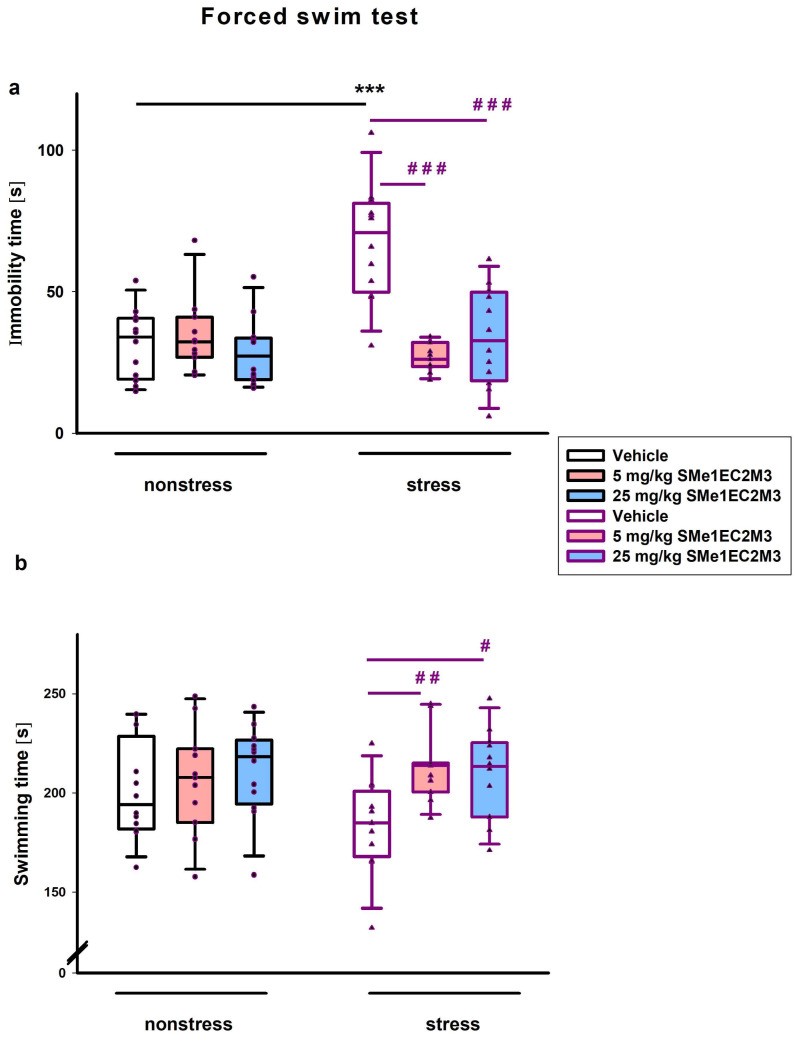
The effect of SMe1EC2M3 on immobility (**a**) and swimming time (**b**) in forced swim test. Means are represented in box plots ± SEM (n = 10–12 animals/group). Two-way ANOVA revealed main effect of treatment (*p* ≤ 0.001), stress vs. nonstress conditions (*p* ≤ 0.01), and their interactions (*p* ≤ 0.001) on immobility time (**a**) and main effect of treatment (*p* ≤ 0.05) on swimming time (**b**). Fisher’s LSD post hoc test revealed significantly different values, marked with *** *p* ≤ 0.001 compared to nonstress vehicle group; marked with ^#^ *p* ≤ 0.05; ^##^ *p* ≤ 0.01; ^###^ *p* ≤ 0.001 compared to stress vehicle group.

**Figure 5 ijms-25-00845-f005:**
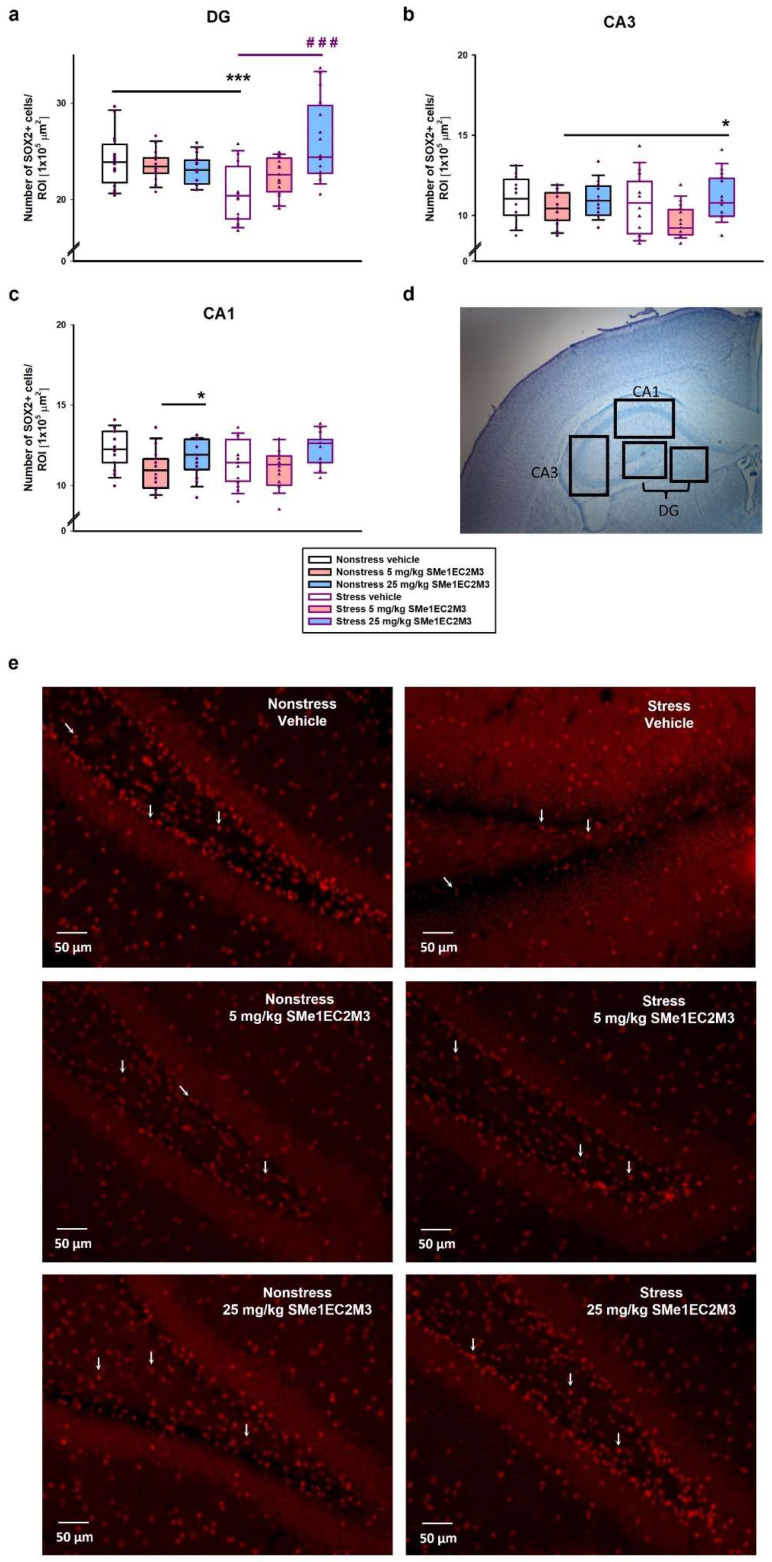
SMe1EC2M3 restores the number of neural progenitor cells (SOX2+) in the hippocampal areas DG (**a**), CA3 (**b**), and CA1 (**c**) in Sprague Dawley rats in a chronic mild stress animal model of depression. Representative image of chosen regions (**d**) and representative microscopic images (20× magnification) showing immunoreactivity to SOX2 (red; white arrows) in the DG (**e**). SOX2+ cells were detected in three different regions of interest (ROIs; 1 × 10^5^ µm^2^) in the DG, CA3, and CA1 parts of the hippocampus (n = 6 animals/group, 8 sections of bilateral hippocampus/group, 3 ROIs/each part of the hippocampus). Means are represented in box plots ± SEM. Two-way ANOVA revealed main effect of the factors interaction (*p* ≤ 0.001) and treatment (*p* ≤ 0.001) for the DG region (**a**) and main effect of treatment for the CA1 and CA3 regions (*p* ≤ 0.01). Tukey–Kramer post hoc revealed significantly different values, marked with * *p* ≤ 0.05; *** *p* ≤ 0.001 compared to nonstress vehicle group; ^###^ *p* ≤ 0.001 compared to stress vehicle group.

**Figure 6 ijms-25-00845-f006:**
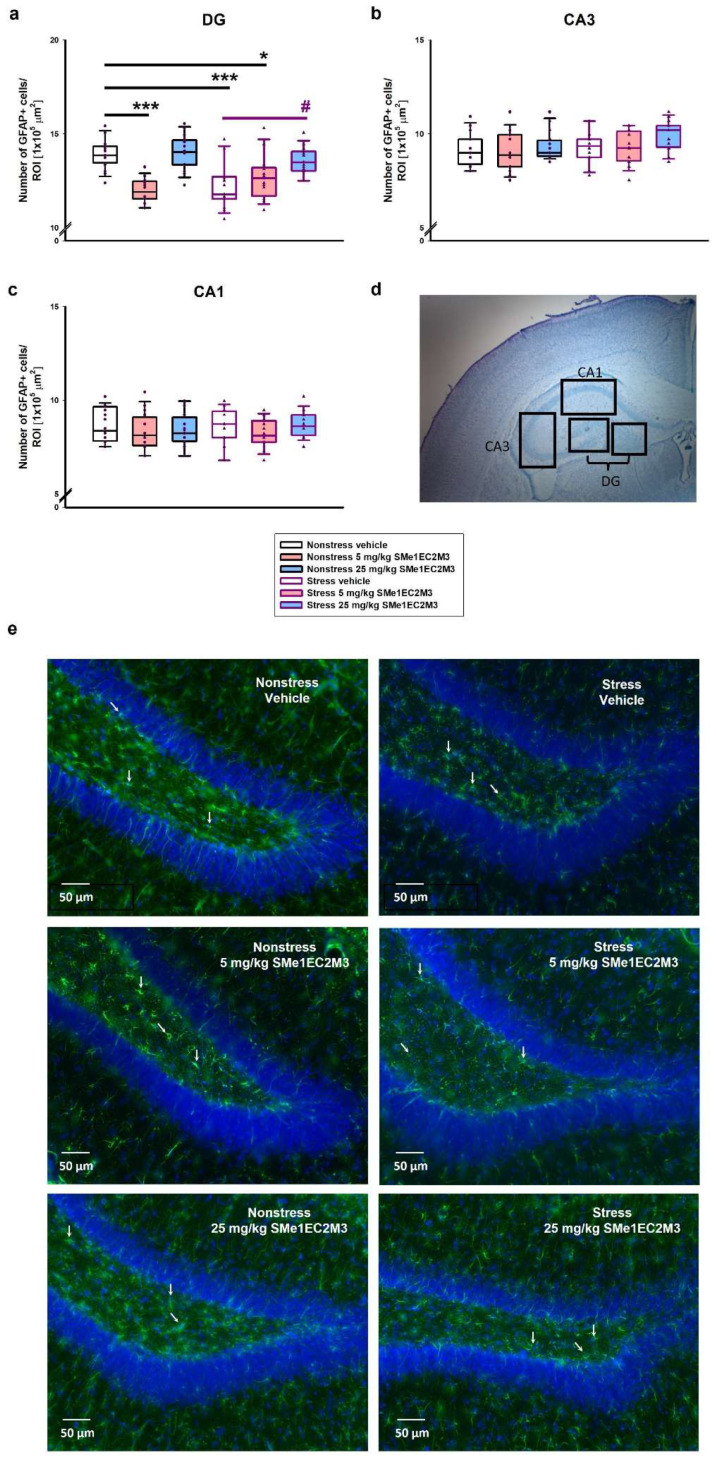
SMe1EC2M3 restores the number of glial cells (GFAP+) in the hippocampal areas DG (**a**), CA3 (**b**), and CA1 (**c**) in Sprague Dawley rats in a chronic mild stress animal model of depression. Representative image of chosen regions (**d**) and representative microscopic images (20× magnification) showing immunoreactivity to GFAP (green; white arrows) and DAPI (blue) in the DG (**e**). GFAP+ cells were detected in three different regions of interest (ROIs; 1 × 10^5^ µm^2^) in the DG, CA3, and CA1 parts of the hippocampus (n = 6 animals/group, 8 sections of bilateral hippocampus/group, 3 ROIs/each part of the hippocampus). Means are represented in box plots ± SEM. Two-way ANOVA revealed main effect of the factors interaction (*p* ≤ 0.001), stress vs. nonstress condition (*p* ≤ 0.05), and treatment (*p* ≤ 0.001) for the DG region. Tukey–Kramer post hoc revealed significantly different values marked with * *p* ≤ 0.05; *** *p* ≤ 0.001 compared to nonstress vehicle group; ^#^ *p* ≤ 0.05 compared to stress vehicle group.

**Figure 7 ijms-25-00845-f007:**
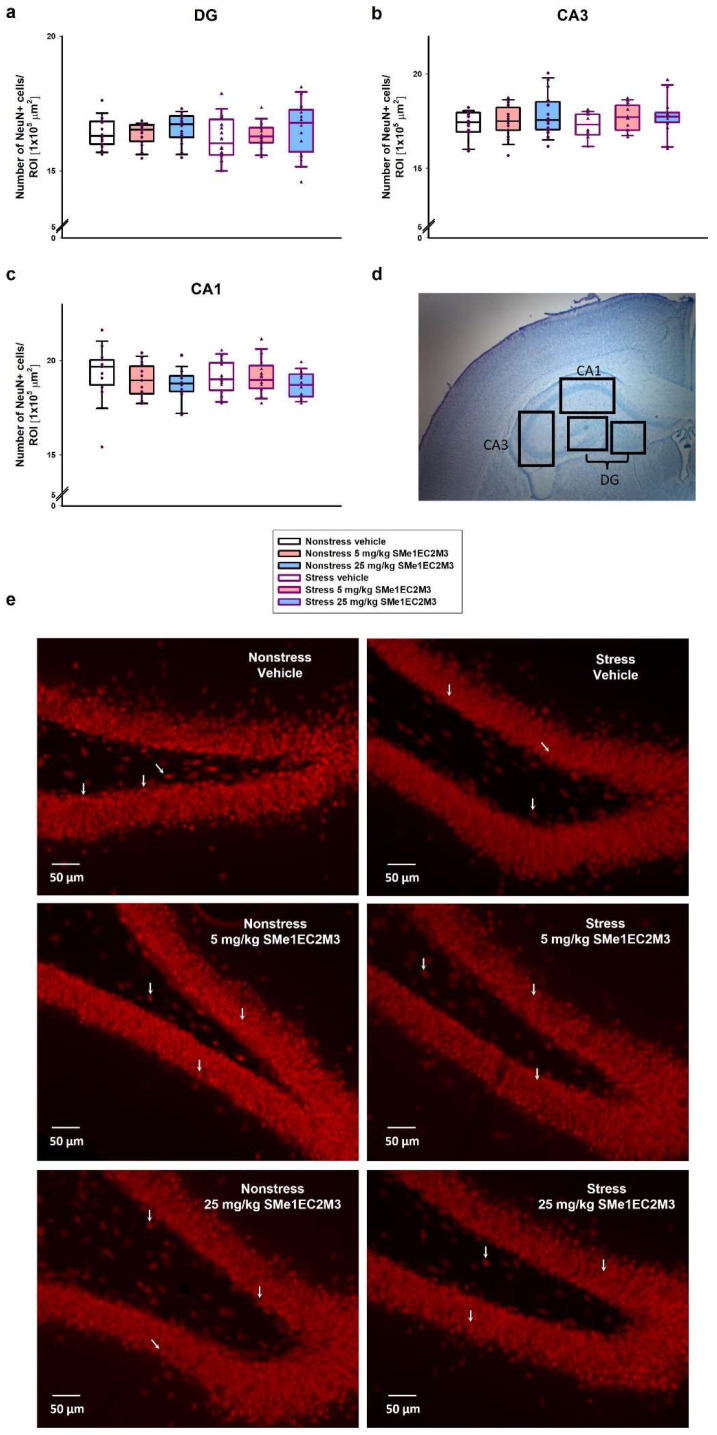
Effect of SMe1EC2M3 on the number of neurons (NeuN+) in the hippocampal areas DG (**a**), CA3 (**b**), and CA1 (**c**) in Sprague Dawley rats in a chronic mild stress animal model of depression. Representative image of chosen regions (**d**) and representative microscopic images (20× magnification) showing immunoreactivity to NeuN (red; white arrows) in the DG (**e**). NeuN+ cells were detected in three different regions of interest (ROIs; 1 × 10^5^ µm^2^) in the DG, CA3, and CA1 parts of the hippocampus (n = 6 animals/group, 8 sections of bilateral hippocampus/group, 3 ROIs/each part of the hippocampus). Means are represented in box plots ± SEM.

**Figure 8 ijms-25-00845-f008:**
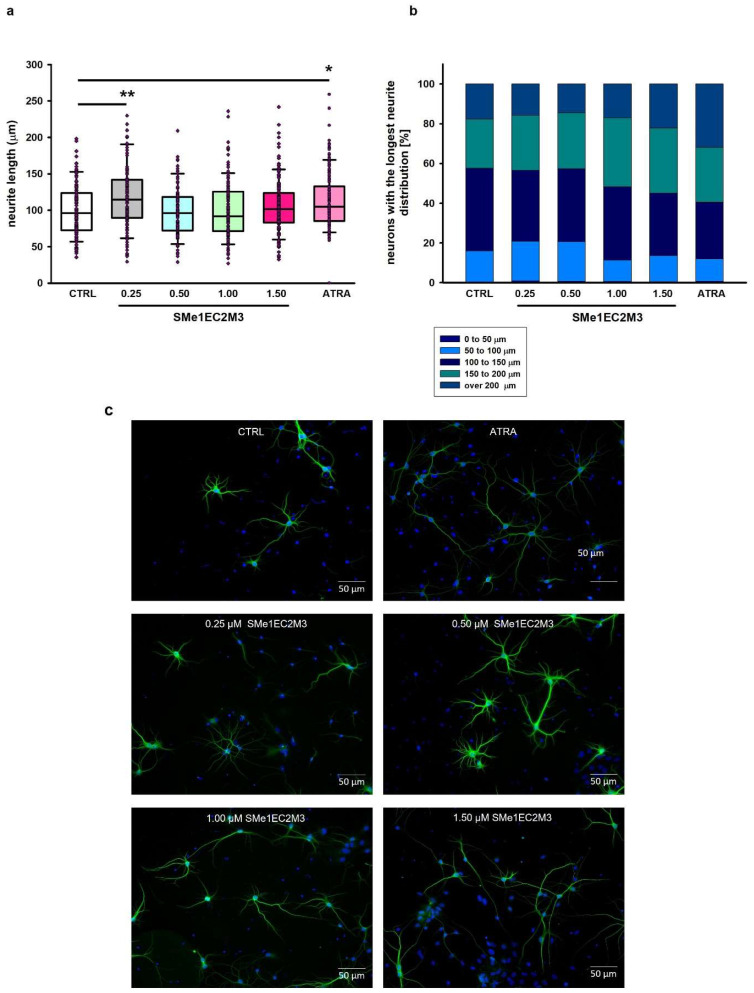
Quantitative assessment of neurite outgrowth in primary hippocampal neurons isolated from P0 Wistar rats in response to different concentrations of SMe1EC2M3 and 10 µM ATRA on DIV9. Neurite length was quantified in all neurons in the visual field from the edge of the nucleus to the apical end of the neurite. Cells were plated at a density of 0.8 × 10^5^/mL (n = 6 pups). Six coverslips per experimental group (n = 100–150 cells) and at least seven areas of interest per coverslip were evaluated. The average value of the longest neurite is shown. Means are represented in box plots ± SEM. ANOVA revealed significant differences between groups. Tukey–Kramer post hoc revealed significantly different values marked with * *p* ≤ 0.05; ** *p* ≤ 0.01 compared to untreated CTRL group (**a**). Stacked bar graphs showing the percentage of neurons with a defined length of neurites (**b**). Representative fluorescent microscopic images of neurons in experimental groups (**c**). Cells labeled MAP2 (green) and nucleus (blue). DIV = day in vitro.

**Figure 9 ijms-25-00845-f009:**
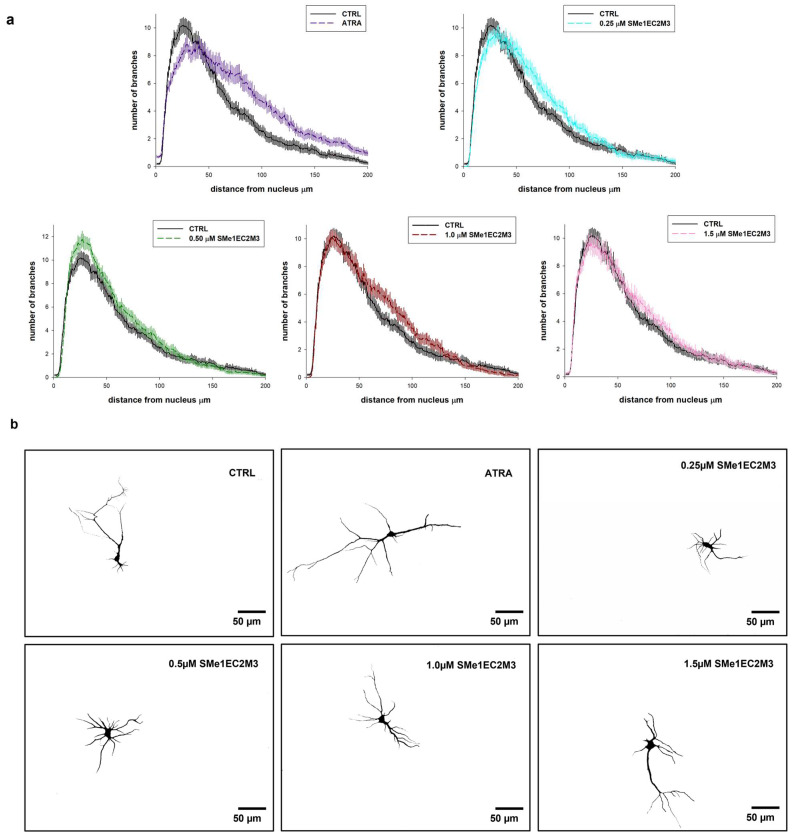
Sholl analysis of neurite arborization in primary hippocampal neurons isolated from P0 Wistar rats in response to different concentrations of SMe1EC2M3 and 10 µM ATRA on DIV9. The number of neurite intersections for concentric circles with various distances far from the nucleus was evaluated using FIJI/IMAGE J. Cells were plated at a density of 0.8 × 10^5^/mL (n = 6 pups) (**a**). Representative binary images of neurons for all experimental groups are shown (n = 40 cells per group) (**b**). Scale bar represents 50 µm. DIV = day in vitro.

**Figure 10 ijms-25-00845-f010:**
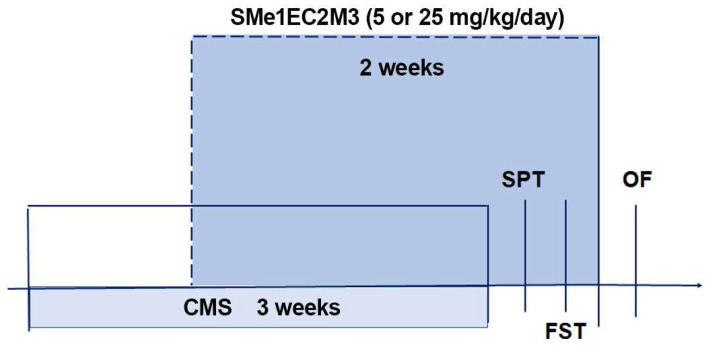
Experimental design. Experimental schedule of chronic mild stress applied for three weeks, antidepressant treatment with SMe1EC2M3 in two doses for two consecutive weeks, and behavioral testing schedule. CMS—chronic mild stress, FST—forced swim test, OF—open field test, SPT—sucrose preference test.

**Table 1 ijms-25-00845-t001:** Schedule of stressor application in chronic mild stress animal model of depression.

Stressor	Days of CMS
overcrowding	2, 13, 17
air puff	4, 8, 19
wet bedding	1, 12, 15, 18
tilted cage	6, 11
predator stress	3, 10, 16
food deprivation	5, 9, 14

**Table 2 ijms-25-00845-t002:** Primary antibodies. ICC—immunocytochemistry; IHC—immunohistochemistry; GFAP—glial fibrillary acidic protein; MAP2—microtubule-associated protein 2; NEUN—neuronal nuclear protein; SOX2—SRY (sex-determining region Y)-box 2.

Name	Host Species	Dilution	Method	Product Number
anti-SOX2	Rabbit	1:500	IHC	AB5603; Merck Millipore, Burlington, MA, USA
anti-NEUN	Rabbit	1:1200	IHC	ab177487; Abcam, Cambridge, UK
anti-GFAP	Mouse	1:500	IHC	G3893; Sigma-Aldrich, Darmstadt, Germany
anti-MAP2	Mouse	1:2000	ICC	M4403; Sigma-Aldrich, Darmstadt, Germany

**Table 3 ijms-25-00845-t003:** Fluorescent secondary antibodies. ICC—immunocytochemistry; IHC—immunohistochemistry.

Name	Host Species	Dilution	Method	Product Number
anti-mouse Alexa Fluor 488	Goat	1:500	IHC/ICC	A-11001; Thermo Fisher Scientific, Bratislava, Slovakia
anti-rabbit Alexa Fluor 555	Goat	1:500	IHC/ICC	A-21429; Thermo Fisher Scientific, Bratislava, Slovakia

## Data Availability

The data that support the findings of this study are available from the corresponding author upon reasonable request.

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
