# Peer review of "Effect of a New Substance with Pyridoindole Structure on Adult Neurogenesis, Shape of Neurons, and Behavioral Outcomes in a Chronic Mild Stress Model in Rats"

_ijms, 2024, doi:10.3390/ijms25020845_

Round 1

Reviewer 1 Report

Comments and Suggestions for Authors

Dear Alexandra Zvoziolova and co-authors, 

I found your study "The effect of the new substance with pyridoindole structure on adult neurogenesis, shape of neurons and behavioral outcomes in chronic mild stress model in rat" interesting with high potential to the translational studies. Overall, I do not have major concerns, but have several suggestions which could improve quality of the manuscript: 

1. if there are some analogs of your experimental compound (SMe1EC2M3), it is worth to discuss, add one paragraph in Introduction. 

2. Also, given that SMe1EC2M3 acts as triple reuptake inhibitor, affecting NE, 5-HT and DA release, it is also worth to add in Discussion one paragraph about efficacy of multitarget Antidepressants vs monotherapy. 

3. Proliferation activity of the compound should be also assessed in the subventricular pathway in parallel to DG hippocampal pathway. 

4. Results: Figure 2 - add a sign for a significant effect of stress on sucrose preference in vehicle-treated rats; 

Lack of SMe1EC2M3 treatment on Sucrose preference might be also related to procedure. Based on own experience with mice, sucrose preference was monitored daily, for a week. It might be useful to record data for several days as well to reduce variability. By the way, please, double-check your raw data, especially for stress group treated with SMe1EC2M3 at highest dose. You may remove outliers and re-analyze data 

 5. Methods:

a. what was an interval between tests? 

b. open field was done in a dim or bright lit condition? Please, change "maze" on "open field" , p. 17. and also "mazes" on p.16 

c. when was compound injected? before or after performance of the sucrose preference test and forced swim test?

Author Response

I found your study "The effect of the new substance with pyridoindole structure on adult neurogenesis, shape of neurons and behavioral outcomes in chronic mild stress model in rat" interesting with high potential to the translational studies. Overall, I do not have major concerns, but have several suggestions which could improve quality of the manuscript:

  1. if there are some analogs of your experimental compound (SMe1EC2M3), it is worth to discuss, add one paragraph in Introduction.

Response:

We thank the reviewer for this comment. We added a paragraph related to the pyridoindole analogs in the introduction section.

  1. Also, given that SMe1EC2M3 acts as triple reuptake inhibitor, affecting NE, 5-HT and DA release, it is also worth to add in Discussion one paragraph about efficacy of multitarget Antidepressants vs monotherapy.

Response:

We thank the reviewer also for this comment. We added paragraph related to multitarget antidepressants that by affecting multiple neurotransmitter systems such as norepinephrine (NE), serotonin (5-HT), and dopamine (DA) release, may potentially enhance efficacy compared to monotherapy. This broader impact on neurotransmitter release allows for modulation across various neural pathways, potentially resulting in a more comprehensive approach to managing depression and related disorders.

  1. Proliferation activity of the compound should be also assessed in the subventricular pathway in parallel to DG hippocampal pathway.

Response:

Although we understand this comment, in the design of this study we evaluated proliferation only in the hippocampus. We added to the manuscript that this represents a limitation of the study.

  1. Results: Figure 2 - add a sign for a significant effect of stress on sucrose preference in vehicle-treated rats;

Response:

We thank the reviewer for this comment. We again looked at the data and re- analyzed them, we excluded outliers and according to that, we added the significances in the graph.

Lack of SMe1EC2M3 treatment on Sucrose preference might be also related to procedure. Based on own experience with mice, sucrose preference was monitored daily, for a week. It might be useful to record data for several days as well to reduce variability. By the way, please, double-check your raw data, especially for stress group treated with SMe1EC2M3 at highest dose. You may remove outliers and re-analyze data

Response:

Following the recommendation of the reviewer we re-analyzed the data. After excluding outliers using GraphPad PRISM’s ROUT method, the results related to the sucrose preference test and forced swim test provided new information (Fig.2, Fig.4).

Methods section:

a. what was an interval between tests?

Response:

We added the interval between tests into the method section 4.3 Behavioral Analysis.

b. open field was done in a dim or bright lit condition? Please, change "maze" on "open field" , p. 17. and also "mazes" on p.16

Response:

We added the light condition interval into the 4.6 open field section. We also corrected and replaced the word "maze".

c. when was compound injected? before or after performance of the sucrose preference test and forced swim test?

Response:

We thank the reviewer for this comment, the compound was injected always at the same time (once daily at 12 am) during the chronic mild stress (CMS) exposure period and afterward during the period of behavioral testing. The compound was injected after the performance to prevent interference between the test and i.p. injection as a possible additional stress factor especially in the chronic scenario of the substance application.

Reviewer 2 Report

Comments and Suggestions for Authors

In the current study, Alexandra Zvozilova and her colleagues investigated the bioactivity of SMe1EC2M3, a synthetic pyridoindole-based compound, in relation to chronic mild stress in rats. The authors conducted an extensive experiment with rats, carefully analyzing the effect of SMe1EC2M3 on the animals' behavioral responses to stress (sucrose preference test, open field test, forced swimming test) and stress-induced disturbances in hippocampal neurogenesis (immunohistochemistry studies), as well as the influence of the tested compounds on neurogenesis in primary neuronal cells. Although the observed changes in response to stress/compound at the tissue and cell level were mostly weak (less than 50%), these changes were statistically significant. Given the demonstrated protective effect of SMe1EC2M3 against chronic mild stress, including restoration of hippocampal neurogenesis, and its ability to enhance neurite outgrowth, this compound can be considered as a drug candidate for the therapy of depressive disorders. This work is undoubtedly novel and interesting and can be published in Int. J. Mol. Sci. after some revisions:

Main comments:

1. In my opinion, it would be good to strengthen somewhat the section devoted to the effect of SMe1EC2M3 in terms of neuritogenesis in primary hippocampal neurons (lines 171-199). Given the journal's focus on molecular studies, it would be interesting to assess the effect of SMe1EC2M3 on key regulators of neuritogenesis in these cells (either at the mRNA or protein level), if the authors are able to do so. This would greatly enhance the article.

2. The SMe1EC2M3 molecule is quite hydrophobic. Dear authors, please provide sufficient evidence that this compound was soluble in water (vehicle) at the doses used. Perhaps the weak effect of the compound at the tissue level was due to the low solubility of the molecule studied.

Minor comments:

p. Page 4, line 124 - in the sentence "main effect t of treatment", please delete t

Line 177, line 503 - what does DIV7 mean?

Line 259, line 270 - what does DIV9 mean? DIV9 is decoded only in line 506. Please correct.

Author Response

Main comments:

  1. In my opinion, it would be good to strengthen somewhat the section devoted to the effect of SMe1EC2M3 in terms of neuritogenesis in primary hippocampal neurons (lines 171-199). Given the journal's focus on molecular studies, it would be interesting to assess the effect of SMe1EC2M3 on key regulators of neuritogenesis in these cells (either at the mRNA or protein level), if the authors are able to do so. This would greatly enhance the article.

Response:

We plan to investigate the mechanisms of neuritogenesis in connection with SMe1EC2M3 effects in the future. As this is the first study in this direction, unfortunately, we don't have such samples in this experimental setting. We added a comment to the discussion.

  1. The SMe1EC2M3 molecule is quite hydrophobic. Dear authors, please provide sufficient evidence that this compound was soluble in water (vehicle) at the doses used. Perhaps the weak effect of the compound at the tissue level was due to the low solubility of the molecule studied.

Response:

We thank the reviewer for this comment. The process of dissolution was performed following a protocol from Koprdova et al. [10]. The substance SMe1EC2M3 was initially crushed into tiny particles and subsequently dissolved in sterile water for injection while being continuously mixed. Stirring improved the dissolution process due to the substance's larger surface area.

Minor comments:

Page 4, line 124 - in the sentence "main effect t of treatment", please delete t

Response:

We corrected the sentence.

Line 177, line 503 - what does DIV7 mean?

Response:

We apologize for the incomplete indication of the abbreviation; we added it to the correct place.

Line 259, line 270 - what does DIV9 mean? DIV9 is decoded only in line 506. Please correct.

Response:

We corrected the abbreviation.

Round 2

Reviewer 2 Report

Comments and Suggestions for Authors

Dear authors, thank you for your responses to my comments and for correcting the manuscript. I ask the authors to try to include mechanical studies in future papers, as this will greatly strengthen your publications. Please pay special attention to the development of soluble SMe1EC2M3 compositions, as the use of suspensions (the mixture of tiny particles of hit compound with sterile water) is highly undesirable for intravenous administration due to the risk of vascular embolism. Otherwise, a very interesting and high quality study. The authors have answered all my questions. The article is ready for publication in Int. J. Mol. Sci.